# Assessment of appropriateness of hospitalisations in Ukraine: analytical framework, method and findings

Feng Zhao,[1] Olena Doroshenko [ID],[2] Valery N Lekhan,[3] Lilia V Kriachkova,[3] Alona Goroshko[4]

[1]HNP, World Bank Group, Washington, District of Columbia, USA
[2]HNP, World Bank Group, Kyiv, Ukraine
[3]Department of Social Medicine and Health Management, Dnipropetrovsk Medical Academy (DMA), Dnipro, Ukraine
[4]Department for Development of Benefits Package, National Health Service of Ukraine, Kyiv, Ukraine

**Correspondence to**
Dr Olena Doroshenko;
inglecasket@gmail.com

## ABSTRACT

**Objectives** This article reviews the applicability of a customised version of the Appropriateness Evaluation Protocol (AEP) to evaluate the magnitude of inappropriate hospitalisations in two regions of Ukraine.

**Data and methods** The original AEP was modified to develop a customised tool, which included criteria for the appropriateness of hospitalisation and duration of inpatient stay. The customisation of the tool followed the Delphi procedure. We randomly selected 381 medical records to test the feasibility and reliability of the method and 800 medical records to evaluate the scope of inappropriate hospitalisations. We used descriptive and analytical statistics, receiver operating characteristic curve analysis and Cohen's kappa to check the consistency between the findings of primary reviewers and experts.

**Result** We observed high levels of agreement in conclusions of primary reviewers (reference standard) and experts during testing of the reliability and validity of the method. The external validity check showed that the use of the tool by different experts provided high accuracy: 95.1 sensitivity, 76.6 specificity and area under ROC-curve (AUC)=0.948 (p<0.001) for analysis of the appropriateness of admissions; 95.3 sensitivity, 84.7 specificity and AUC=0.900 (p=0.001) for the duration of hospitalisations. Cohen's kappa coefficient ($\kappa$) indicated agreement in expert evaluations of 0.915 (95% CI 0.799 to 1.000) and 0.812 (95% CI 0.749 to 0.875), respectively. We found that over one-third of admissions (38.1%; 95% CI 33.9 to 43.5) and over half of total bed-days were unnecessary (57.4%; 95% CI 56.4 to 58.5). The highest levels of stay were observed in hospitals' general medicine departments (64.6%; 95% CI 63.0 to 66.3) compared with other departments included in the analysis.

**Conclusion** The proposed method is robust in assessing the appropriateness of hospitalisations and duration of inpatient stays. The quantified levels of unnecessary hospital care indicate the need for improving efficiency and quality of care and optimising the excessive hospital capacities in Ukraine.

## INTRODUCTION

Hospital care is a core part of any national health system; it influences health outcomes of the population and requires significant funding.[1 2] Still, the use of hospital care has distinct features across the globe and ranges

## Strengths and limitations of this study

- ► An objective evaluation of inpatient care is possible through the application of a unified method for the assessment of the appropriateness of hospital admissions and duration of hospitalisations, which is customised for Ukraine and is based on clear criteria for inpatient treatment.
- ► The received data on the scope of inappropriate hospitalisations and their excessive duration provide a reliable base to estimate the needs of Ukraine's population in hospital care, which is especially relevant for the current reform of the health sector in the country.
- ► We evaluated the appropriateness of hospitalisations but did not evaluate the actual need in hospital care in Ukraine, and the capacities of healthcare institutions to adequately address the needs.
- ► Because electronic medical records were not available, the quality of the assessment of paper-based medical records was compromised by the structure and legibility of handwritten text.
- ► The future use of the proposed method in practice may require additional training for evaluators; its sustainability will depend on the uptake of the method at the national and at facility levels.

in terms of the level of effectiveness.[3] If inpatient care fails to improve the health outcome when compared with less invasive or less intensive non-hospital care, it is considered an overuse of inpatient services.[4–6]

Overuse of hospital care is typical for many countries around the world, ranging from 1% to 54% in general,[5] and up to 90% for some diseases.[7] Unnecessary hospitalisations are a matter of concern for many nations, specifically due to the rising trend of healthcare expenditures and a significant proportion of hospital expenditures in overall health spending.[8–12]

Hospital sector expenditures are mostly dependent on admission processes, length of patient hospital stays and advancement of medical technologies in hospital facilities.

Inpatient length of stay is the most important contributing factor influencing the use of resources and cost of care during hospitalisation; it closely relates to the effectiveness of hospital treatment, the effectiveness of hospital management and general efficiency of a health system.[13–15]

Rationalisation of hospital expenditures together with increased, or at least sustained, quality of hospital care are key objectives of healthcare reform in Ukraine.[15 16] The hospital sector in Ukraine is oversized and inefficient by both the numbers of beds and hospitals. The number of beds per 10 000 people was 74.3 in 2016, or three times higher than in the European Union (EU); the number of hospitals per 10 000 people exceeded EU numbers by a quarter; the average length of stay was 11.2 days, or 3 days longer than for patients in the EU.[17]

Numerous studies published in Ukraine within the last 15 years have been dedicated to the use of hospital care,[18–24] and the research has shown significant inefficiencies in the hospital sector. Reported results, however, vary substantially: between 8.2% and 70.6% inappropriate hospitalisations, and 26.4% to 86.5% of unnecessary bed-days.

Studies in other countries have reported similar problems with the variability of results, which led to an international recommendation from researchers that a unified method for the assessment of reasons for hospitalisations and their duration should be used,[7 8] specifically the Appropriateness Evaluation Protocol (AEP).[25 26] The AEP has high validity and reliability compared with other methods for assessment of inappropriate hospitalisations[2 27] and was customised for many countries.[7 8 28]

The AEP method has not been used in Ukraine, where assessments are traditionally based on expert evaluations providing retrospective analyses of medical records. Such analyses are typically designed by individual experts through methods that are not validated.[18–24] Although the provision of hospital care is guided by legislation and clinical protocols, concrete and evidence-based criteria for hospitalisations are absent. Additionally, a more accurate estimate of the needs in hospital care is necessary to support the ongoing health reform in Ukraine, which is implemented to improve access and quality to healthcare, and to address inefficiencies in service delivery.

**The objective** of this study was to develop and test the customised Ukraine version of the unified method for assessment of the appropriateness of hospitalisations, including reasons for admissions to hospitals and duration of hospital stays, and to evaluate the scope of inappropriate hospitalisation of adults in Ukraine using a sample of hospitals from two regions.

## METHODS
### Ethics statement
Confidentiality of information from medical records was kept by protecting patient identifiers. Researchers received copies of paper-based records with patient identification fields crossed out or covered. Names of physicians and hospital names were replaced with ID codes for reviewers of medical records.

All experts working with primary data signed non-disclosure agreements of confidential data and obligations to follow ethics principles of medical research statements, which are endorsed by international agreements and the Ministry of Health of Ukraine. Compliance with bioethics principles and medical deontology was confirmed in the conclusion of the Committee on bioethics and medical deontology in the State Institution 'Dnipropetrovsk Medical Academy of the MOH Ukraine' (Protocol No. 6 of 16 September, 2016).

### Patient and public involvement
We did not involve patients or the public in our work.

### Tool development
Based on available international tools[8 9 25–30] and existing approaches used in Ukraine,[18–24] the tool for this study was developed and customised for the Ukrainian context in order to assess the appropriateness of hospitalisations and duration of inpatient stays. The tool included criteria for appropriateness of hospital admissions and a guide for expert evaluations of justified inpatient lengths of stay. The tool was finalised using the Delphi method for expert evaluations[31] in three rounds as schematically presented in figure 1. To avoid the potential conflict of interests, experts represented different healthcare facilities, provided informed consent, received only summarised information of inputs provided before each round and did not communicate about the study before the final approval of the modifications to the AEP. The group of reviewers controlled the adherence to the Delphi procedure. Criteria for hospital admissions were included in the customised tool based on an analysis of seven versions of the AEP: original US version (US-AEP),[25] European (E-AEP),[26] Turkish (TR-AEP),[29] Dutch (D-AEP),[28] German (DE-AEP),[30] Chinese (C-AEP)[9] and Iranian version of the AEP (IR-AEP).[8] The customised Ukrainian AEP tool defined Ukraine-specific indications for hospital admission and criteria for the justification of inpatient length of stay — summarised in the Ukrainian Appropriateness Evaluation Protocol (UA-AEP) (see online supplementary file 1).

We used criteria from these versions of the UA-AEP to develop a guide that helps to extract medical information about hospital care episodes from medical records (MoH form 003/o) for evaluation of the appropriateness of admission decisions and justification of lengths of stay (see online supplementary file 2).

### Piloting of the method
The proposed tool was piloted in three types of hospitals of different size in one of the largest regions in eastern Ukraine. Four department profiles were selected for the analysis: general medicine, cardiology, neurology and surgery. These departments are representing a most

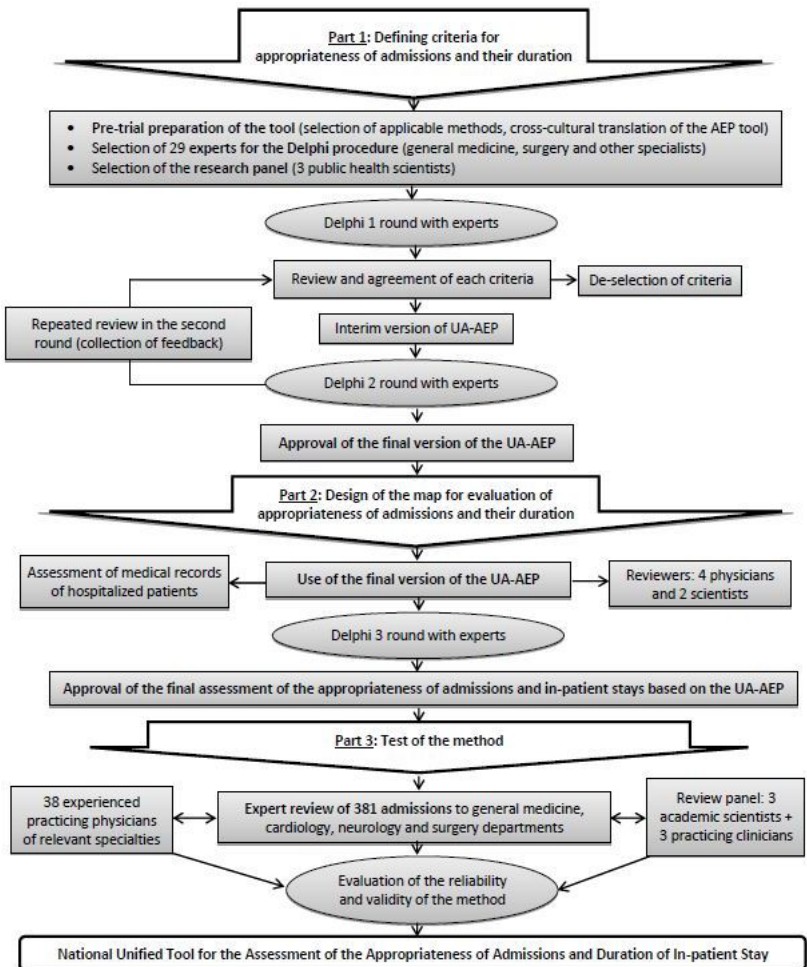

**Figure 1** Schematic presentation of the steps followed during the development and evaluation of the UA-AEP method.

typical structure of a hospital: on average, 44% of all adult patients are admitted to these departments in Ukraine.

Standard medical records were used for analysis (MoH approved Form 003/o). The two-stage approach was used to sample records for the year preceding the analysis: first, 1 month was randomly selected (March 2016); second, from all archived records in the four departments in March 2016, 381 cases were randomly selected. All sampled records were proportionally distributed for evaluation to 38 experts with relevant clinical backgrounds for the profile of sampled cases. Experts were selected based on their area of expertise and the results of the initial test of competencies. The total number of experts was determined based on an average anticipated workload of 10 medical records per one expert.

The results of the piloting confirmed the feasibility of the tool for data collection and analysis. The pilot data showed that about one-third of all admissions included in the analysis were classified as inappropriate. The length of stay in facilities was often excessive: 45.2% of all bed-days were inappropriate (see online supplementary table 1). The yielded results corresponded with the finding from a study of inappropriate hospitalisations in the same region,[20] and this confirmed the feasibility of the proposed method.

### Reliability and validity of the method

The results of the pilot were verified by a review panel, consisting of three primary reviewers (public health scientists). Each primary reviewer was assisted by a clinician in the relevant clinical field (therapist, surgeon and neurologist).

We assessed the reliability and validity of the method by comparing the evaluation results provided by 38 experts with the conclusions of the review panel. The latter was used as a reference standard for the conclusions.

### Statistical methods

The data set and statistics were prepared using Microsoft Excel (Microsoft Office 2016 Professional Plus, Open License 67528927) and STATISTICA 6.1 software (StatSoft Inc, serial No. AGAR909E415822FA). We used descriptive and analytical statistics for data analysis.[32] For relative values, 95% CI were calculated based on the corrected Wald method. The assessment of the validity in the differences in relative indices was carried out according to Pearson's $X^2$ test.

For the sample calculations, we used Cochran formula with p=0.95 (α=0.05) and t=1.96. The general number of all hospital admissions N for catchment area served by the hospitals included in pilot sites was 755 350 and the

hospital admission rate 23.3%, making the calculation for the minimal powered sample of 275 admissions for the pilot study (sampled 381). For the main stage of the analysis, the recommended sample of 322 records (α=0.05) or 895 records (α=0.03) was calculated using 'Power analysis' in STATISTICA 6.1 software for the general number of admissions in Ukraine of 8 594 210, and the level of inappropriate hospitalisations of 29.9% calculated for the pilot (sampled 800).

To analyse agreement in experts' reviews, we used the concordance coefficient (W) and Cronbach's alpha (α). Approximation of coefficient values to 1 indicates an increase in agreement rates.

The inter-rater agreement in experts' reviews was analysed using overall agreement coefficient and Cohen's kappa coefficient (κ), applying the following ranges: values of 0.61 to 0.80 were defined as good strength of agreement, values of 0.81 and higher as very good strength of agreement.[32]

ROC curve analysis was conducted to calculate operational characteristics and the area under ROC-curve to evaluate the discriminative capacity of the method. The relationship between the area under the ROC curve and diagnostic accuracy was assessed using the following ranges: 0.7 to 0.8 – good, 0.8 to 0.9 – very good, 0.9 to 1.0 – excellent.[33]

The statistical significance was taken at the level of $p<0.05$ (5%) for all tests.

## RESULTS
### Results of reliability and validity tests
The Ukraine version of the AEP proved its reliability and validity for the assessment of the hospitalisations because sensitivity and specificity measures are high for the sample and within the selected medical specialities for evaluation of the appropriateness of admissions and duration of inpatient stay (table 1).

Using Cohen's kappa (κ), we observed a high level of agreement in experts' reviews. Results for agreement on the appropriateness of admissions was 0.915 (95% CI 0.799 to 1.000) and duration of inpatient stay was 0.812 (95% CI 0.749 to 0.875).

The proposed method has very high operational characteristics: the area under the ROC curve only slightly varies for different medical specialities and averages to 0.948 for the appropriateness of admissions and to 0.900 for the duration of stay, which defines the potential to properly distinguish different patterns (discriminative capacity) of the method as excellent.

### Core sample
Admissions at inpatient departments of secondary and tertiary care facilities were studied in two regions of Ukraine (central and western). The total of 12 healthcare facilities was included in the study, of which nine facilities are secondary care hospitals and three are tertiary care facilities.

The expert assessment covered a random sample of 800 medical records in four medical specialities: general medicine, neurology, surgery and cardiology (online supplementary table 2). A similar approach was used to sample records for the core sample as in the pilot. All medical records for patients discharged during the month preceding the beginning of data collection (July 2016) were collected from all departments of the general medicine, surgery, neurology and cardiology profile. The

**Table 1** Inter-rater agreement of experts' evaluations and operational characteristics of the method for assessment of the appropriateness of hospitalisations

| Statistics | General medicine n=100 | Cardiology n=70 | Neurology n=119 | Surgery n=92 | Total pilot sample n=381 |
|---|---|---|---|---|---|
| **Appropriateness of admissions** | | | | | |
| Sensitivity (%) | 84.8 | 96.3 | 96.4 | 97.3 | 95.1 |
| Specificity (%) | 73.1 | 87.5 | 68.5 | 94.7 | 76.6 |
| Area under ROC curve (p value) | 0.865 (0.002) | 0.977 (<0.001) | 0.945 (<0.001) | 0.991 (<0.001) | 0.948 (<0.001) |
| Overall agreement (%) | 84.0 | 94.2 | 88.2 | 90.5 | 92.8 |
| Cohen's κ (95% CI) | 0.702 (0.566 to 0.839) | 0.916 (0.871 to 0.961) | 0.888 (0.781 to 0.996) | 0.934 (0.843 to 1.000) | 0.915 (0.799 to 1.000) |
| **Duration of inpatient stay** | | | | | |
| Sensitivity (%) | 88.4 | 96.3 | 96.6 | 97.2 | 95.3 |
| Specificity (%) | 85.9 | 87.5 | 75.0 | 94.4 | 84.7 |
| Area under ROC curve (p value) | 0.870 (0.009) | 0.920 (<0.001) | 0.860 (0.007) | 0.960 (<0.001) | 0.900 (0.001) |
| Overall agreement (%) | 86.7 | 91.9 | 90.1 | 94.3 | 91.2 |
| Cohen's κ (95% CI) | 0.737 (0.604 to 0.870) | 0.838 (0.684 to 0.992) | 0.772 (0.639 to 0.905) | 0.902 (0.794 to 1.000) | 0.812 (0.749 to 0.875) |

ROC, receiver operating characteristic.

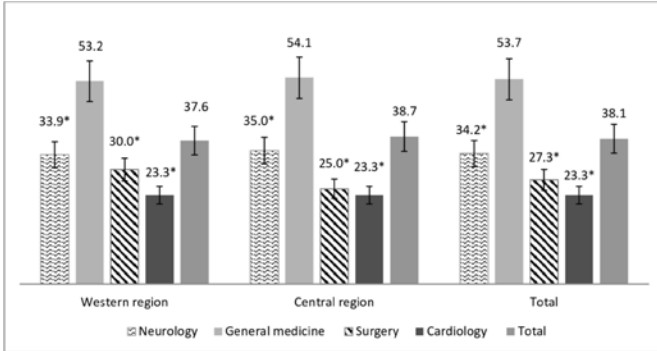

**Figure 2** Shares of inappropriate admissions by department type, total and in the two regions of Ukraine (%, 95% CI).*p<0.001 compared with the general medicine department.

total number of sample medical records made up 8% to 12% of the total number of patients discharged from the selected departments during that month.

Experts, according to their specialisation, reviewed medical records following the agreed guide for the analysis of the appropriateness of hospitalisations.

### Results of data analysis for two regions of Ukraine

The findings of the expert review show that over one-third of cases — 38.1% (95% CI 33.9 to 43.5) in the central region, 38.7% (95% CI 34.8 to 41.5) in the western region and 37.6% (95% CI 32.8 to 42.3) in total were inappropriate hospital admissions (figure 2). Every seventh patient (13.3%; 95%CI 6.5 to 20.8%) admitted to an inpatient department of tertiary care facilities could have been more appropriately treated at secondary care facilities with adequate equipping and staffing.

In our sample, over half of the total bed-days spent by patients in hospitals were classified as inappropriate or longer than necessary (57.4%). The volume of unnecessary bed-days in the western region was far greater (63.1%) than in the central region (51.7%, p<0.001), which was related to a higher rate of inappropriate admissions at secondary care facilities of patients to all specialised departments except the neurology profile (table 2).

The analysis of appropriateness of hospital admissions by specialities showed that the highest level of inappropriate admissions was recorded for patients treated in general medicine departments (53.7%); lower levels (p<0.001) were identified for all the other specialities: neurology – 34.2%, surgery – 27.3% and cardiology – 23.3%.

The highest share of unnecessary stay was determined for patients of general medicine departments (64.6%), and the lowest share in cardiology (51.2%); for the neurology and surgery departments these shares were also high at 54.6% and 52.4% of total bed-days, respectively.

The experts were asked to specify one of the proposed reasons for inappropriate hospitalisation in each medical record reviewed. The following statistics describe the major reasons of inappropriate hospital admissions and unnecessary inpatient stays: to keep patients under care

in the absence of other facilities with capacity to provide care to patients after discharge (21.0%; 95% CI 15.1 to 26.9), and to treat patients with overestimated severity of their condition (20.4%; 95% CI 14.6 to 26.3) (see online supplementary table 3 for all reasons).

### DISCUSSION

Like in the other countries,[8 9 28–30] we have customised the European version of the AEP to adapt it to Ukrainian specifics. The scope of changes is marginal: we have amended some criteria, included relevant criteria and modified the wording. Such changes of the AEP are explained by the lower technological intensity of healthcare provision in Ukraine and peculiarities of service delivery in the country. The proposed Ukrainian version of Appropriateness Evaluation Protocol is a reliable and valid tool for assessment of the appropriateness of hospitalisations for Ukraine's hospitals, as confirmed by the high levels of Cohen's kappa coefficient ($\kappa$) for agreement on criteria of appropriateness and duration of hospitalisations (0.915 and 0.812, respectively).

Our study generates two key results regarding hospital care in Ukraine. First, there is a need for accurate assessment of the appropriateness of hospital admissions. To increase the objectivity of assessing the use of hospital resources, we suggest applying a unified method, customised for Ukraine, which gives reliable results for analysing the appropriateness and duration of hospital admissions. Second, the reserves for optimisation of the hospital sector in Ukraine are high, as evidenced by the volume of inappropriate hospitalisations in the two regions estimated using the proposed method.

There are several limitations that may limit the generalisability of the findings related to possible seasonal variations for hospital care statistics and selection of regions that may not be fully representative for the whole of Ukraine. To minimise and adjust for limitations, we compared the sample data from two regions with the data collected from an additional region during the pilot stage. We did not find any significant differences between the regional statistics: the level of inappropriate admissions in the pilot was 30.7% (95% CI 26.0 to 35.4), which is slightly lower than the total for the surveyed regions but could be explained by the inclusion of municipal hospitals in the pilot, which had lower levels of inappropriate hospitalisations in the main sample as well. We, therefore, assume that the facilities in our research are representative of inpatient facilities in Ukraine.

Among other limitations we report that we did not evaluate the actual need in hospital care in Ukraine, and the capacities of healthcare institutions to adequately address the needs. Because electronic medical records were not available, the quality of the assessment of paper-based medical records was compromised by the structure and legibility of handwritten text. We also suggest that the future use of the proposed method in practice may require additional training for evaluators; its sustainability

**Table 2** Volumes of inappropriate hospitalisations

| Characteristic of the sample | Number of inappropriate admissions | Total bed-days for inappropriate hospitalisations | Average length of stay in bed-days (95% CI) | Unnecessary days of inpatient stay, % (95% CI) |
|---|---|---|---|---|
| Total for two regions | | | | |
| By facility level | | | | |
| Secondary care | 217 | 2255 | 7.1 (6.9 to 7.2) | 58.8 (57.5 to 60.1) |
| Tertiary care | 88 | 1274 | 8.2 (7.8 to 8.5) | 54.8 (52.9 to 56.6)* |
| Both levels | 305 | 3530 | 7.4 (7.3 to 7.5) | 57.4 (56.4 to 58.5) |
| By speciality of departments | | | | |
| Neurology | 79 | 1123 | 7.9 (7.7 to 8.0) | 54.6 (52.6 to 56.5)† |
| General medicine | 151 | 1083 | 8.8 (8.6 to 9.0) | 64.6 (63.0 to 66.3) |
| Surgery | 54 | 838 | 5.9 (5.7 to 6.0) | 52.4 (50 to 54.7)† |
| Cardiology | 21 | 486 | 7.3 (7.0 to 7.5) | 51.2 (48.1 to 54.3)† |
| Central region | | | | |
| By facility level | | | | |
| Secondary care | 105 | 1252 | 8.2 (8 to 8.4) | 50.6 (48.7 to 52.6)‡ |
| Tertiary care | 49 | 740 | 8.5 (8.3 to 8.7) | 53.4 (51.0 to 55.9) |
| Both levels | 154 | 1993 | 8.3 (8.2 to 8.4) | 51.7 (50.1 to 53.2)‡ |
| By speciality of departments | | | | |
| Neurology | 79 | 1123 | 7.9 (7.7 to 8.0) | 55.0 (51.2 to 58.8) |
| General medicine | 151 | 1083 | 8.8 (8.6 to 9.0) | 58.4 (56.1 to 60.7)‡ |
| Surgery | 54 | 838 | 5.9 (5.7 to 6.0) | 41.9 (38.9 to 44.9)†‡ |
| Cardiology | 21 | 486 | 7.3 (7.0 to 7.5) | 46.6 (42.9 to 50.2)†‡ |
| Western region | | | | |
| By facility level | | | | |
| Secondary care | 112 | 1002 | 6.0 (5.7 to 6.3) | 65.9 (64.2 to 67.6) |
| Tertiary care | 39 | 533 | 7.7 (6.6 to 8.8) | 56.6 (53.8 to 59.4)* |
| Both levels | 151 | 1535 | 6.5 (6.3 to 6.7) | 63.1 (61.7 to 64.6) |
| By speciality of departments | | | | |
| Neurology | 58 | 830 | 7.8 (7.5 to 8.2) | 54.4 (52.1 to 56.7)† |
| General medicine | 59 | 366 | 7.8 (7.3 to 8.3) | 72.7 (70.3 to 75.1) |
| Surgery | 27 | 237 | 3.8 (3.4 to 4.3) | 67.4 (63.9 to 70.8) |
| Cardiology | 7 | 102 | 4.9 (4.1 to 5.6) | 63.0 (57.3 to 68.7)† |

*p≤0.001 compared with the secondary level facilities.
†p<0.001 compared with the general medicine departments.
‡p<0.001 compared with the other region.

will depend on uptake of the method at the national and at facility levels.

The results of our study point to a high volume of inappropriate hospitalisations and their unjustified duration in Ukraine. The number of unnecessary bed-days for appropriately hospitalised patients exceeded the number of bed-days from inappropriate admissions to the hospital by 27.8% (95% CI 24.4 to 31.1). This means that appropriately hospitalised patients often stay in the hospital longer than necessary. This share is especially large for patients hospitalised in general medicine departments — the numbers of unnecessary bed-days for appropriately admitted patients exceeds the number of bed-days for inappropriately admitted patients by 37.0% (95% CI 33.7 to 40.2).

We asked experts engaged in the evaluations to propose financial, economical and organisational measures to prevent inappropriate hospitalisations or excessively long inpatient treatment (online supplementary table 4). Close to a quarter of respondents (24.3%; 95% CI 18.1 to 30.6) proposed to eliminate the normative regulation specifying the number of staff based on the number of beds in the facility. This principle inherited from the Soviet era, incentivised hospitals to maintain a large number of beds regardless of the actual needs for inpatient care. Although the

issue had been included in the reform agenda for several decades,[15] it was only addressed in 2016 when the Ministry of Health abolished the existing normative regulation. This change, however, has not yet resulted in significant optimisation of hospital capacities.

Hospitals in Ukraine do not differentiate by the intensity of interventions. In most hospitals, both acute and chronic patients, as well as those who need nursing and social care, can stay in the same departments to receive medical care. Another proposed measure for improvement of hospital efficiency was to right-size the hospital network to align it with actual need in hospital care and categorise capacity of hospitals by intensity of care (14.9% of respondents, 95% CI 9.7 to 20.1), as well as the use of clear indications for admission to inpatient departments for facilities of different levels (32.6%, 95% CI 25.8 to 39.4). Such measures were proposed at the preliminary stage of the health reforms (2010 to 2014)[15] but not yet implemented.

The most popular measure proposed by experts was the introduction of alternatives to inpatient care: the replacement of inpatient services with outpatient services provided at the primary care level and the organisation of treatment services without overnight stay (44.8% of respondents, 95% CI 37.5 to 52.0). This suggestion is in line with the current global trend of changes in the health system organisation.[34 35] Currently, the avoidance of hospitalisations of patients with ambulatory care-sensitive conditions (ACSCs) is included as a special policy direction on the agenda of many countries.[36–38] The rate of the appropriateness of hospitalisations can be seen as an indicator of the quality of primary healthcare.[39 40] Larger capacity of primary care to treat ACSCs results in a significant reduction of the hospitalisations of patients with these conditions.[41] Daycare clinics and daycare centres, as well as socially appropriate and less expensive forms of medical care, are widely used in different countries as alternatives to hospital services.[3 42 43]

To increase the availability of alternative care for patients from remote areas, an organisation of daycare coupled with discharge to a boarding house facility has proven to be effective.[44] In such facilities, a patient, after receiving the relevant service or intervention, is placed in a boarding house where a range of services is available, with a capacity for the provision of necessary care in the case of complications. This approach was favoured by 33.7% of respondents (95% CI 26.8 to 40.6).[44]

Daycare facilities are also gaining popularity in Ukraine. In 2016, 31.6% more patients received medical care in daycare facilities compared with the total number of hospitalised patients, but the increase in the use of daycare facilities did not replace inpatient care.[15] According to large-scale international studies, the combination of incentives and special training of medical staff can help to improve the use of daycare instead of hospitalisations.[45 46]

An additional measure is the timely discharge of patients from hospitals (proposed by 19.9% of experts, 95% CI 14.1 to 25.7). The issue of appropriate discharge is not unique to Ukraine. European experts point out that there are obstacles to timely discharge in multiple countries.[1]

Interestingly, experts did not consider better coordination with social services as one of the factors to reduce inappropriate hospitalisations and duration of stay (only 5% respondents agreed with this statement, 95% CI 1.7 to 8.2), which can be explained by the low level of social service development and weak coordination between health and social sectors in the country. At the same time, strong evidence supports the need to create an interface between health and social care providers, which can help reduce levels of hospitalisations and their duration for patients, especially elderly.[47 48]

The current health reform prepared and launched by the Government and the Ministry of Health in 2018 focuses primarily on changing the payment mechanisms, which will influence changes in the hospital sector. At the same time, reforms also cover the primary healthcare and emergency services to ensure accessibility of services within the scope of limited available resources in the country. The results received and future use of the proposed method for the assessment of the appropriateness of hospitalisations can inform decisions in hospital restructuring.

Optimisation of inpatient care by reducing inappropriate hospitalisations and excessive lengths of stay in hospitals, given the current economic situation, may become more than just an important direction in the development of the healthcare system to increase effectiveness. It can essentially help to sustain good progress in the improvement of health sector organisation and health outcomes by adequately meeting the needs of the population in hospital care. Although the suggested transformations of the hospital sector may require additional funds for structural reorganisation, training and motivation of medical staff, these investments are likely to be offset by the large efficiency gains from hospital optimisation.

**Acknowledgements** The authors thank all staff working in Health Departments of the two regions who facilitated the process of data collection, assisted with the preparation of the primary data set and participated in the discussion for data interpretation.

**Contributors** FZ - Critical review of the article for important intellectual content, OD - Drafting the article, preparing the first concept of the study, VL - Substantial contribution to the concept and design of the study, overall guidance and coordination of the study, LK - Substantial contribution to the acquisition of data, analysis and interpretation of data, and the first draft of the article, AG - Contributions to the first draft of the article.

**Funding** The study was supported by the World Bank (within Improving effectiveness in human development and social accountability Project financed by the Department for International Development).

**Competing interests** None declared.

**Patient consent for publication** Not required.

**Provenance and peer review** Not commissioned; externally peer reviewed.

**Data availability statement** Data are available in a public, open access repository at http://datadryad.org/ with the doi: 10.5061/dryad.2v6wwpzh9

and indication of whether changes were made. See: https://creativecommons.org/licenses/by/4.0/.

**ORCID iD**
Olena Doroshenko http://orcid.org/0000-0002-0741-2863

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
