## [Reviewer comments · BMJ Open]

ARTICLE DETAILS

TITLE (PROVISIONAL)	Assessment of appropriateness of hospitalizations in Ukraine: analytical framework, method, and findings
AUTHORS	Zhao, Feng; Doroshenko, Olena; Lekhan, Valery; Kriachkova, Lilia; Goroshko, Alona

VERSION 1 – REVIEW

REVIEWER	Lorenzo Paglione Sapienza - Università di Roma
REVIEW RETURNED	20-May-2019

GENERAL COMMENTS	Overall, the article presents interesting and relevant findings of the functioning of the health service in Ukraine. A need for an improvement of the hospital system of the country emerged, but this work cannot be defined as conclusive in terms of policy definition. A necessary and more context-specific evaluation, firstly of the health needs of the population is needed in order to make decisions on the future of the Ukrainian hospital service. Title: Assessment of avoidable hospitalizations in Ukraine: analytical framework, method, and findings The work is about the development of a customized tool for evaluating the appropriateness of hospital stay, based on the Appropriateness Evaluation Protocol. In the introduction is explained how Length of Hospital Stay (LOS) influence health expenditure, giving an overview of the hospital service in Ukraine and of the international tool. In methods is summarily described the procedure for the definition of the customized tool. The pilot study is well described, both in methods and in results, with significant findings of the appropriateness of hospital stay in a large number of Ukrainian Hospital, covering the different contexts of the country. The sample of department and patient is well defined. Statistical analysis and results are accurately presented, but some results are not presented in tables (in particular the share of inappropriate admissions). The discussion is largely coherent with results, limitations have to be better defined. Major revisions Authors have to clarify the difference between the term "avoidable" and "appropriate". In the text, these two terms are both used for describing the same thing, but they have two different meanings. The developed tool (Appropriateness Evaluation Protocol) is about the appropriateness of admissions and hospital stay, and the method used for developing it is correct and valid, but in the first and in the last part of the text authors have to distinguish the use of the terms. I think the more correct term should be
--

	"appropriateness", and authors have to avoid the use of "avoidable", especially in the title and in the introduction. The difference between these two terms have to be explained in the discussion, where ACSCs are cited. ACSCs, in fact, are about avoidable, but appropriate, hospitalizations. Authors should define what "therapy department" means, is unclear in the text if this definition refers to a specialistic diagnostic and therapy department or to an intensive care department. I suggest to refer to MeSH terms to better standardize a definition that is context-dependent. Minor revisions  - Authors should make explicit the disclosure of Conflict of Interests in participants to the DELPHI procedure for defining the customized tool (UA-AEP). - In the introduction, at the end of part 3 – Problem Description, a reference is needed to justify that the length of stay (LOS) is the most contributing factor influencing the use of resources. Reference n. 14 is too old, and, in this article, the same proposition is based on a study of 1976. - In the discussion, in my opinion, the importance of social work have to be underlined. There is interesting literature about how hospital social workers can reduce the length of stay and hospital readmissions when are caused by social problems that may have an impact on health status.
--	---

REVIEWER	Cristina Canova University of Padova, Padova
REVIEW RETURNED	10-Jun-2019

GENERAL COMMENTS	This is quite an interesting paper on the assessment of avoidable hospitalization developing a customized version of the AEP. The authors should better argue the main contribution of this study on the international knowledge. Furthermore the text is often verbose and not flowing. The introduction is a bit too long. Particularly the 4th-7th paragraphs include many local details that may be summarized and integrated in max two paragraphs. Is it not clear whether health service in Ukraine is universalistic or what? The Methods section includes many results of the pilot study that should be summarized. Statistical methods are presented in the 'reliability and validity...' and the same methods are furthermore presented in 'statistical methods'. In the Results 'core sample' section a lot of methodological details are presented. Minor Changes Page 2 11 "scope of unnecessary hospitalizations" could be clarified. Page 8 12 "pilot sites sites was 755,350" what does this refer to? Figure 1 maybe could be structured in a clearer manner, especially part 1 and 2.
---

REVIEWER	Maria Kompoti Intensive Care Unit, Thriassio General Hospital of Eleusis, Athens, Greece
REVIEW RETURNED	16-Jul-2019

GENERAL COMMENTS	This is a very interesting study investigating the rate of unnecessary hospital admissions in Ukraine. The authors used developed a tool based on the Delphi procedure. The proposed tool was implemented in three hospitals differing in size and type and in different department types, ensuring a global overview of the tool's discriminating ability. I have no major concerns about the study adequacy in design, conduction and analysis. Just a minor remark: in table 1, I would substitute "parameters" for "statistics", since statistics of certain hospital samples are provided and not the parameters of the entire Ukrainian hospitalized population.
--

VERSION 1 – AUTHOR RESPONSE

Comments	Responses
Reviewer: 1 (Lorenzo Paglione, Institution and Country: Sapienza - Università di Roma)	
Overall, the article presents interesting and relevant findings of the functioning of the health service in Ukraine. A need for an improvement of the hospital system of the country emerged, but this work cannot be defined as conclusive in terms of policy definition. A necessary and more context-specific evaluation, firstly of the health needs of the population is needed in order to make decisions on the future of the Ukrainian hospital service.	Thank you for finding our article interesting and relevant. We agree that the assessment of the health needs of population is necessary to make decisions on the future of the Ukrainian hospital service. We included such statement before line 41 on page 5 of the original manuscript and as a limitation of our study (line 24 on page 3).
Reviewer: 2 (Cristina Canova, Institution and Country: University of Padova, Padova)	
This is quite an interesting paper on the assessment of avoidable hospitalization developing a customized version of the AEP. The authors should better argument the main contribution of this study on the international knowledge. Furthermore the text is often verbose and not flowing.	Thank you for this comment. We included an additional paragraph to the discussion to explain the contribution of this study to the international knowledge on AEP application (line 56 on page 11).
The introduction is a bit too long. Particularly the 4th-7th paragraphs include many local details that may be summarized and integrated in max two paragraphs. Is it not clear whether health service in Ukraine is universalistic or what?	Thank you for advising to reduce the level of details provided. We keep only essential information from paragraphs 4-7.
The Methods section includes many results of the pilot study that should be summarized. Statistical methods are presented in the ‘reliability and validity...’ and the same method are furthermore presented in ‘statistical methods’	Thank you for pointing out to duplications. We have deleted repetitive information and summarized methods as possible.
In the Results ‘core sample’ section a lot of methodological details are presented	Accepted: unnecessary information excluded from the description of the core sample (see line 49 on page 9).
Minor Changes Page 2 11 “scope of unnecessary hospitalizations” could be clarified. Page 8 12 “pilot sites sites was 755,350” what does this refer to?	Thank you for the attention to details: Page 2-11: we changed unnecessary to inappropriate. We use “inappropriate” in the same meaning as in the AEP Page 8-12: we clarified that the number referred to the general number of all hospital

Figure 1 maybe could be structured in a clearer manner, especially part 1 and 2.	admissions N for catchment area served by the hospitals included in the pilot sites Figure 1: amended
Reviewer: 3 (Maria Kompoti, Intensive Care Unit, Thriassio General Hospital of Eleusis, Athens, Greece)	
This is a very interesting study investigating the rate of unnecessary hospital admissions in Ukraine. The authors used developed a tool based on the Delphi procedure. The proposed tool was implemented in three hospitals differing in size and type and in different department types, ensuring a global overview of the tool's discriminating ability. I have no major concerns about the study adequacy in design, conduction and analysis.	Thank you for your interest to the study and acceptance of the study's adequacy in design, conduction and analysis
Just a minor remark: in table 1, I would substitute "parameters" for "statistics", since statistics of certain hospital samples are provided and not the parameters of the entire Ukranian hospitalized population	Accepted, in table 1 "parameters" are changed for "statistics"
Additional comments in the .docx file	
The work is about the development of a customized tool for evaluating the appropriateness of hospital stay, based on the Appropriateness Evaluation Protocol. In the introduction is explained how Lenght of Hospital Stay (LOS) influence health expenditure, giving an overview of the hospital service in Ukraine and of the international tool. In methods is summarily described the procedure for the definition of the customized tool. The pilot study is well described, both in methods and in results, with significant findings of the appropriateness of hospital stay in a large number of Ukrainian Hospital, covering the different contexts of the country. The sample of department and patient is well defined. Statistical analysis and results are accurately presented, but some results are not presented in tables (in particular the share of inappropriate admissions). The discussion is largely coherent with results, limitations have to be better defined.	Thank you for the overall positive assessment. As to the specific comment, we would like to clarify the following: the share of inappropriate admissions is presented in figure 2, and we decided to not include the repetitive information to the table On limitations: as suggested by one of the reviewers, we explained that we only assessed the actual hospitalizations without assessing the existing needs. This is included as the limitation of the study (line 24 on page 3).
Major revisions Authors have to clarify the difference between the term "avoidable" and "appropriate". In the text, these two terms are both used for describing the same thing, but they have two different meanings. The developed tool (Appropriateness Evaluation Protocol) is about the appropriateness of admissions and hospital stay, and the method used for developing it is correct and valid, but in the first and in the last part of the text authors have to distinguish the use of the terms. I think the more correct term should be "appropriateness", and authors have to avoid the use of "avoidable", especially in the title and in the introduction. The difference between these two terms have to be explained in the discussion, were ACSCs are cited.	Thank you for emphasizing this inconsistency. We fixed the terminology, using "appropriate" and "inappropriate" to be consistent with the terminology of the AEP. On ACSC, we agree that we could have assessed the avoidable, but appropriate, hospitalizations, but this was not the focus of our study. We hope that ACSC will be studied in detail for Ukraine in future research. We have replaced "therapy departments" with "general medicine" departments.

ACSCs, in fact, are about avoidable, but appropriate, hospitalizations. Authors should define what "therapy department" means, is unclear in the text if this definition refers to a specialistic diagnostic and therapy department or to an intensive care department. I suggest to refers to MeSH terms to better standardize a definition that is context-dependent.	
Minor revisions  - Authors should make explicit the disclosure of Conflict of Interests in participants to the DELPHI procedure for defining the customized tool (UA-AEP). - In the introduction, at the end of part 3 – Problem Description, a reference is needed to justify that the length of stay (LOS) is the most contributing factor influencing the use of resources. Reference n. 14 is too old, and, in this article, the same proposition is based on a study of 1976.  - In the discussion, in my opinion, the importance of social work have to be underlined. There is interesting literature about how hospital social workers can reduce the length of stay and hospital readmissions when are caused by social problems that may have an impact on health status. 	We explained how we addressed the conflict of interests for participants to the DELPHI procedure for defining the customized tool (UA-AEP) in “Methods” section (line 28 on page 6). Thank you for pointing to the reference No.14. We have replaced the old source with the reference to the newer publication: Khosravizadeh O., Vatankhah S., Bastani P., Kalhor R., Alirezaei S., Doosty, F. Factors affecting length of stay in teaching hospitals of a middle-income country. Electron Physician. 2016 Oct; 8(10): 3042–3047. Published online 2016 Oct 25. doi: 10.19082/3042 We agree that the importance of social work needs to be highlighted. We included an additional paragraph to the discussion to explain the attitude and potential importance of social care for the reduction of inappropriate hospitalizations (see line 60 on page 13).

VERSION 2 – REVIEW

REVIEWER	Lorenzo Paglione Department of Epidemiology-Lazio Regional Health Service, ASL Roma 1, Rome, Italy
REVIEW RETURNED	01-Oct-2019
GENERAL COMMENTS	The authors have lightened the conclusions regarding the need to reduce the availability of beds, updated the bibliography and definitely enriched the discussion, considering the many aspects that concern the process of contact and access to the health services. Limits are defined and commented, and the conclusions are in line with the results presented and discussed. Only a small clarification would perhaps be required regarding the ACSCs: the sentence concerning the appropriateness is partly ambiguous because these Ambulatory Care-Sensitive Conditions concern hospitalization avoidable by primary care but appropriate once conditions occur.